# Valence band engineering of GaAsBi for low noise avalanche photodiodes

Yuchen Liu [1], Xin Yi [1,2], Nicholas J. Bailey [1], Zhize Zhou[1,3], Thomas B. O. Rockett[1], Leh W. Lim[1], Chee H. Tan[1], Robert D. Richards [1] & John P. R. David [1 ✉]

Avalanche Photodiodes (APDs) are key semiconductor components that amplify weak optical signals via the impact ionization process, but this process' stochastic nature introduces 'excess' noise, limiting the useful signal to noise ratio (or sensitivity) that is practically achievable. The APD material's electron and hole ionization coefficients ($\alpha$ and $\beta$ respectively) are critical parameters in this regard, with very disparate values of $\alpha$ and $\beta$ necessary to minimize this excess noise. Here, the analysis of thirteen complementary p-i-n/n-i-p diodes shows that alloying GaAs with $\leq 5.1\%$ Bi dramatically reduces $\beta$ while leaving $\alpha$ virtually unchanged—enabling a 2 to 100-fold enhancement of the GaAs $\alpha/\beta$ ratio while extending the wavelength beyond 1.1 μm. Such a dramatic change in only $\beta$ is unseen in any other dilute alloy and is attributed to the Bi-induced increase of the spin-orbit splitting energy ($\Delta so$). Valence band engineering in this way offers an attractive route to enable low noise semiconductor APDs to be developed.

[1] Department of Electronic and Electrical Engineering, University of Sheffield, Sheffield, UK. [2] Present address: Institute of Photonics and Quantum Sciences, School of Engineering and Physical Sciences, Heriot-Watt University, Edinburgh, UK. [3] Present address: State Key Laboratory of CAD&CG, Zhejiang University, Hangzhou, Zhejiang Province, People's Republic of China. ✉email: j.p.david@sheffield.ac.uk

II−V-based semiconductor avalanche photodiodes (APDs) are widely used in optical communication networks because the avalanche gain can increase the sensitivity of the system without sacrificing speed[1,2]. There is, however, a limit to the maximum gain that can be usefully utilized as two factors start to degrade the performance at higher gain values. Firstly, the stochastic nature of the impact ionization process results in 'excess' noise, which increases with increasing gain, limiting the maximum sensitivity[3]. Secondly, APDs have a gain-bandwidth product (GBP) due to the multiple transits of the high field region necessary to achieve the gain, limiting the maximum speed that can be achieved[4]. Both these factors depend critically on the $\alpha/\beta$ ratio of the semiconductor material that is used for the APD, and to obtain the lowest excess noise and the highest speed, the carrier with the larger ionization coefficient must initiate the avalanche multiplication process. Silicon APDs are very widely used in a range of applications as the $\alpha/\beta$ ratio is fairly large over a wide range of electric fields[5] and this can yield fairly low excess noise. Silicon, however, has an indirect bandgap energy ($Eg$) of 1.1 eV, limiting it to the detection of photons below ~1000 nm. To get sufficient absorption and to operate at the electric fields that give very low excess noise requires very thick, reach-through structures that require very high voltages (> 300 V) to operate[6]. Other semiconductor materials like HgCdTe[7] and InAs[8] have almost infinitely large $\alpha/\beta$ ratios and are capable of much longer wavelength detection; however, their small $Eg$ (~0.35–0.4 eV) means that they need to be cooled to reduce their large dark currents. Monte Carlo simulations show that in most III−V semiconductors capable of room temperature operation (i.e., $Eg \geq 0.7$ eV), both electrons and holes undergo significant scattering prior to an ionization event and, despite differences in the conduction and valence band structures, the $\alpha$ and $\beta$ values are broadly similar in magnitude[9–11], leading to large excess noise and hence limiting their sensitivity. There has therefore been considerable effort directed at developing novel III−V materials and structures that are capable of providing a large $\alpha/\beta$ ratio and of operating with minimal cooling. Changing the alloy composition of a III−V semiconductor normally results in both $\alpha$ and $\beta$ changing by approximately similar amounts; consequently, the $\alpha/\beta$ ratio hardly changes, as seen in InGaAsP[12], AlGaAs[13] or even strained-layer InGaAs/GaAs superlattice material systems[14]. Numerous attempts have been made to enhance the $\alpha/\beta$ ratio using band structure engineering techniques such as heterojunctions[15–17], or by using quantum dots[18,19], but these require increased complexity in the growth and the results to date have been mixed with only the work by Ren et al.[16] showing some enhancement in $\alpha$ from the conduction band-edge discontinuity. While most of these band structure engineering efforts have looked to enhance $\alpha$ while leaving $\beta$ unchanged, more recently there have been reports of how digital alloy growth of some materials may form mini-bands, resulting in the preferential reduction of $\beta$[20,21], and low excess noise. The central difficulty with all these ideas is that carriers capable of ionization are not at the minima of the conduction and valence bands of most semiconductors but are in higher-lying bands at significantly higher energies when electron or hole ionization is initiated[9,11]. Significantly enhancing the $\alpha/\beta$ ratio, therefore, requires the modification of the higher-lying band structure rather than only the lowest conduction and valence band edges, as most attempts have hitherto focused on.

In this paper, we increase the $\alpha/\beta$ ratio in GaAs by suppressing its hole impact ionization through modification of the valence band structure. Bismuth (Bi) is one of the largest atoms that can be incorporated into GaAs. The strong difference in electronegativity between it and the arsenic (As) atoms it replaces causes the Bi to act as an isovalent impurity in GaAs, strongly perturbing the valence band-structure. This leads to not only a significant narrowing of the bandgap via a band anticrossing interaction[22], but more importantly for our interests, an increase in the valence band spin-orbit splitting energy ($\Delta so$) (Fig. 1a–c).

The increase of $\Delta so$ in GaAsBi is expected to reduce Auger recombination in lasers;[23] however, it is difficult to incorporate the high levels of Bi (> 10 %) necessary for this benefit to be demonstrated without deleteriously affecting the material quality due to the low miscibility of GaBi and GaAs. This low miscibility results in a reconstruction-sensitive Bi incorporation mechanism[24,25] that necessitates low crystal growth temperatures[26] and near stoichiometric As/Ga flux ratios;[27] It is, however, possible to produce high-quality opto-electronic devices containing GaAsBi with up to at least 6 % Bi;[28] by this fraction, the valence band structure is already strongly modified. Any increase in $\Delta so$ from the value of 0.34 eV in GaAs should make it harder for holes to scatter from the heavy hole (HH) and light hole (LH) bands into the split-off (SO) band from which their ionization threshold is gained[11], potentially making this parameter instrumental in reducing $\beta$. We show here that introducing small atomic percentages of Bi into GaAs reduces $\beta$ much more than $\alpha$, significantly increasing the $\alpha/\beta$ ratio with a consequent reduction in avalanche excess noise.

Adding Bi during the growth of GaAs results in some As atoms being replaced, as shown in Fig. 1a. This reduces the energy of the bandgap mainly by increasing the valence band energy level, as shown schematically in Fig. 1b, with the consequence that the energy separation between the split-off band and the valence band, $\Delta so$, increases with increasing Bi. Theoretical calculations of the energy levels using a 12-band k•p model and nearest neighbour $sp^3s^*$ tight-binding Hamiltonian as a function of Bi[29] (see Fig. 1c) show that hybridization of the GaAs valence band edge with the resonant, Bi-induced localized states causes the valence band to rapidly increase in energy when a small Bi fraction is added to GaAs. This is accompanied by roughly linear movements of the conduction band (−28.2 meV/%Bi) and split-off band (−5.5 meV/%Bi). These movements result in $\Delta so$ increasing approximately linearly at a rate of 46.7 meV/%Bi for alloy compositions between 1 and 11 % Bi.

The diodes analyzed in this experiment were a series of GaAs based p-i-n and n-i-p structures, grown with intrinsic regions comprising 0–5.1 % Bi and with thicknesses of 200–1600 nm as detailed in Table 1. The structures were grown on n- and p-GaAs (001) substrates using an Omicron scanning tunnelling microscope-molecular beam epitaxy (MBE) system as detailed in the "Methods" section. The compressive strain in some of the thicker layers exceeds the Matthews−Blakeslee critical layer limit[30] with the formation of dislocations partially relaxing the strain; however, the impact of this relaxation on $\Delta so$ (for low Bi content) is minimal as shown in Fig. 1c. The generic structure is shown in Fig. 1d and details of the fabrication of the circular mesa diode structures are given in the "Methods" section. The layer thicknesses and background doping levels were determined using capacitance−voltage measurements. The actual thicknesses were fairly close to the nominal values, as shown in Table 1, and the background doping was found to be < $3 \times 10^{15}$ cm$^{-3}$ in all the layers, suggesting that the electric field across the depletion regions can be assumed to be fairly constant. Determination of the Bi content was undertaken using X-ray diffraction (XRD) and photocurrent measurements. Reciprocal space maps of the 004 and 224 reflections were undertaken on some of the partly relaxed samples to uniquely define their Bi content and relaxation. Absorption coefficients extracted from the photocurrent measurements (Fig. 1e) show that adding even modest amounts of Bi extends the wavelength cut-off of GaAs, with just 3.5 % Bi increasing its absorption coefficient at 1064 nm to more than one hundred times that of silicon. From the

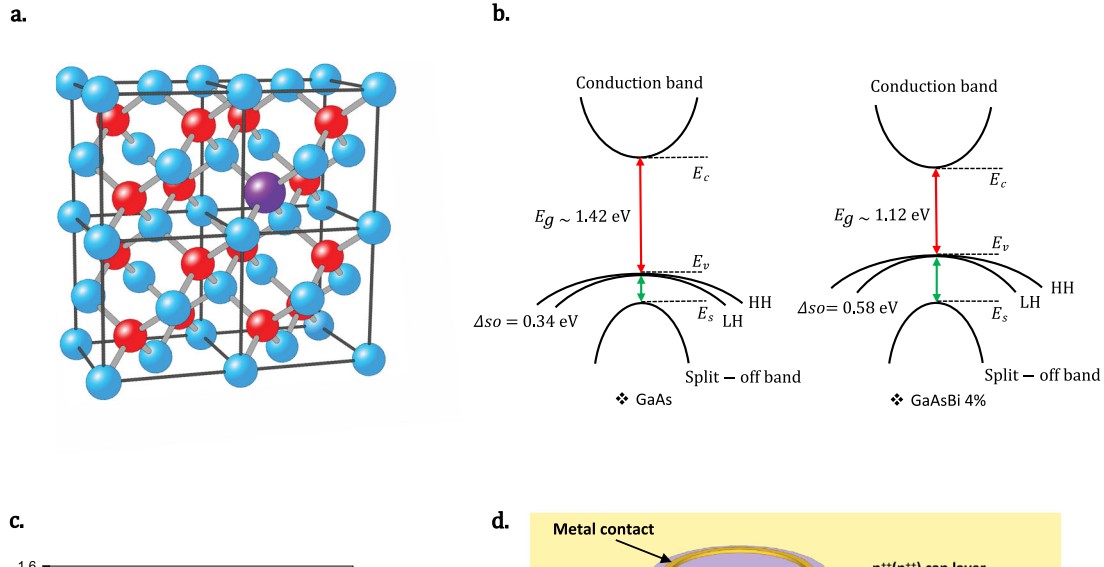

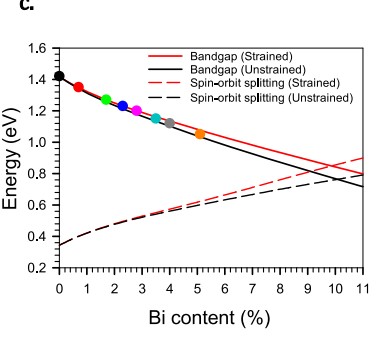

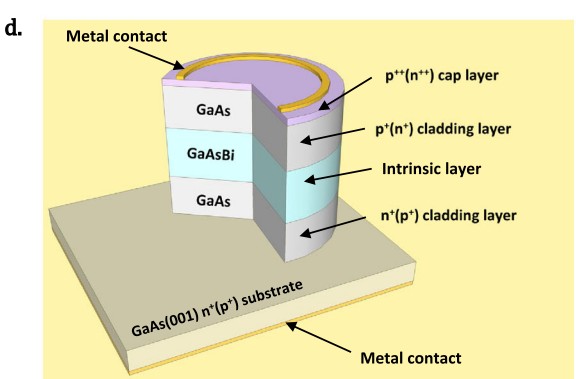

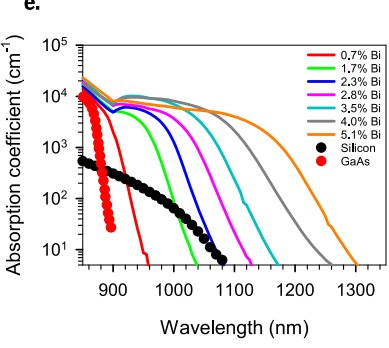

**Fig. 1 Bi incorporation in GaAs and its effect on the opto-electronic properties. a** A schematic showing Bi (purple) replacing As (red) in GaAs. **b** A simplified diagram of the effect of 4 % Bi on the energy bands of GaAs. **c** The bandgap energy and spin-orbit splitting energy of free-standing and strained GaAsBi calculated by Broderick et al[29]. The coloured dots represent the different Bi content samples characterized in this paper; data is also shown in Table 1. **d** Schematic cross-section of the p-i-n (n-i-p) device structures used in this investigation. **e** Absorption coefficient vs wavelength for a range of Bi contents compared to silicon[49] and GaAs[50].

absorption coefficient, the direct bandgap can be obtained by using the following expression:[31]

$$a(h\omega) \propto (h\omega - E_g)^{1/2} \qquad (1)$$

where $h$ is Planck's constant and $\omega$ is the frequency of incident photons. The device's bandgap can therefore be estimated by plotting the square of the absorption coefficient against incident photon energy. Further details of these calculations are provided in the Supplementary section. Plotting $E_g$ obtained this way against the Bi content obtained from the XRD measurements in Fig. 1c shows excellent agreement with the theoretical predictions.

All the samples investigated showed good forward diode characteristics which scaled with device area with ideality factors of ~2. Despite the presence of some dislocations caused by strain relaxation in the thicker structures, low dark currents of < 10 μA were seen in 50–100 μm diameter devices before avalanche breakdown occurred, enabling us to determine their photomultiplication characteristics. A mixture of bulk and surface leakage is thought to be responsible for the reverse dark currents measured in these devices.

Electron and hole initiated photomultiplication ($M_e$ and $M_h$ respectively) measurements were undertaken as functions of reverse bias on these samples as described in the "Methods"

**Table 1 Experimental layer details.**

| Diode type | Layer number | Nominal intrinsic region thickness (nm) | Actual intrinsic region thickness (nm) ± 10 nm | Bi content (%) ± 0.1 % | Band gap (eV) ± 5 meV |
|---|---|---|---|---|---|
| p-i-n | P1 | 200 | 220 | 3.5 | 1.17 |
|  | P2 | 400 | 409 | 4.0 | 1.12 |
|  | P3 | 800 | 747 | 4.0 | 1.12 |
|  | P4 | 1600 | 1420 | 4.0 | 1.12 |
|  | P5 | 400 | 490 | 0 | 1.42 |
|  | P6 | 400 | 450 | 2.3 | 1.23 |
| n-i-p | N1 | 200 | 200 | 3.5 | 1.17 |
|  | N2 | 400 | 377 | 3.5 | 1.17 |
|  | N3 | 800 | 772 | 2.8 | 1.21 |
|  | N4 | 400 | 390 | 0 | 1.42 |
|  | N5 | 400 | 406 | 0.7 | 1.35 |
|  | N6 | 400 | 415 | 1.7 | 1.27 |
|  | N7 | 200 | 207 | 5.1 | 1.05 |

GaAsBi p-i-n and n-i-p layer details. The exact intrinsic region doping levels and thicknesses were determined using capacitance–voltage measurements, while the Bi content and band gap were determined by X-ray diffraction (XRD) and photocurrent measurements.

section, and the results obtained on the GaAsBi p-i-ns (n-i-ps) with 2.8–4.0 % Bi are shown in Fig. 2a, b together with their forward and reverse dark currents. The $M_e$ and $M_h$ measurements undertaken on 50–400 μm diameter devices corroborate the dark current breakdown voltages seen in all the structures. Fig. 2c, d shows how, in nominally 400 nm thick p-i-n (n-i-p) devices, the electron (hole) initiated multiplication varies with increasing Bi content. The data plotted as log $(M_{e,h}-1)$, shows that the measurable onset of the ionization process (defined here as when $M_{e,h} = 1.01$) is only very weakly dependent on Bi content at an electric field of around 210 kV/cm for $M_e$, but that even a small amount of Bi significantly increases the threshold electric field necessary for $M_h$ to occur from 217 kV/cm for GaAs to 333 kV/cm for the 5.1 % Bi structure. This contrasts with, for example, the addition of aluminium to GaAs, where the electric fields necessary for electron and hole ionization both increase by similar amounts with increasing aluminium[32]. The thickness of the avalanche region width does not affect the threshold field for hole ionization to occur per se as shown by the identical values of 290 kV/cm for N2 (377 nm thick and 3.5 % Bi) and N1 (200 nm thick and 3.5 % Bi) in Fig. 2d—only the Bi content has this effect. Using multiplication data from the devices, the ionization coefficients for the different Bi compositions were extracted by solving the ionization integral across the multiplication region[33] given by:

$$M(x_o) = \frac{\exp[-\int_{x_o}^{W}(\alpha - \beta)dx]}{1 - \int_0^W \alpha \exp[-\int_0^x (\alpha - \beta)dx']dx} \quad (2a)$$

where $M(x_o)$ is the multiplication due to the injection of an electron–hole pair at position $x_o$, between the high field region 0 to $W$. In the case of p-i-n or n-i-p structures where a constant electric field can be assumed to exist between 0 to $W$, and only pure electrons or holes initiate the multiplication, this can be simplified to:

$$M_e = \frac{1}{1 - \frac{\alpha}{\beta - \alpha}\{\exp[(\beta - \alpha)W] - 1\}} \quad (2b)$$

$$M_h = \frac{1}{1 - \frac{\beta}{\alpha - \beta}\{\exp[(\alpha - \beta)W] - 1\}} \quad (2c)$$

Fig. 2e, f show these ionization coefficients for a range of GaAsBi containing samples over a wide range of electric fields. The methodology used to extract this data (detailed in the "Methods" section) assumes that $\alpha$ and $\beta$ are functions of the

electric field (following the Chynoweth expression[34]) and the Bi content, and that we can interpolate for Bi contents that are not experimentally available. In this analysis, the effect of any 'dead-space'[35] (the minimum distance carriers need to travel before reaching equilibrium with the electric field) on the multiplication has been ignored. This dead-space was found to reduce the multiplication only when the avalanching width was ≤ 0.1 μm[36] and so can be ignored in these structures. The validity of these ionization coefficients is demonstrated by the simulated $M_e$ and $M_h$ values for the structures (solid lines in Fig. 2c, d), replicating the measured data almost exactly over two orders of magnitude. While the $\alpha$ value only decreases by about a third from that of GaAs as the Bi content increases (see inset of Fig. 2e), the $\beta$ value decreases by orders of magnitude at lower electric fields. Such highly dissimilar changes in ionization coefficients with alloy composition have not been seen in any other material system and appear to be uniquely related to the presence of Bi.

The impact of the Bi on the minimum electric field necessary for ionization to occur (defined as when $M_{e,h} = 1.01$ from Fig. 2b, c) is shown in Fig. 3a. The threshold electric field for electron ionization actually increases slightly with increasing Bi content despite the decreasing bandgap energy, while that for hole ionization increases very rapidly and correlates with increasing $\Delta so$ energy. While the addition of Bi decreases the lowest conduction band (Γ) energy in GaAs, the subsidiary satellite valleys at the X and L points of the Brillouin zone show an increasing energy separation from Γ[37]. Moreover, the ionization transition rate in GaAs is largely dominated by the second conduction band[9] and both of these effects may explain the relatively modest changes seen in $\alpha$ with Bi content. Holes undergoing impact ionization in GaAs, however, gain their threshold energy primarily from the split-off band[11], as the ionization transition rates from the HH and LH bands are much lower than the phonon scattering rates. Furthermore, the HH and LH transition rates decrease abruptly at higher energies due to the termination of these bands at the Brillouin zone edge. Any increase in $\Delta so$ will make it harder for holes to transfer to the SO band and so will reduce the ionization rate at a given electric field. The significance of the $\Delta so$ energy for the $\alpha/\beta$ ratio at higher electric fields corresponding to $M_e = 1.1$ in 400 nm thick GaAsBi p-i-ns can be clearly seen in Fig. 3b. The variation of the $\alpha/\beta$ ratio with both Bi content and electric field (taken from Fig. 2e, f) is shown in Fig. 3c. Although extremely large ratios are seen at the lowest electric fields, practical devices like APDs tend to operate at higher electric fields where the $\alpha/\beta$ ratio is reduced, so we also show the $\alpha/\beta$ ratios (symbols) which

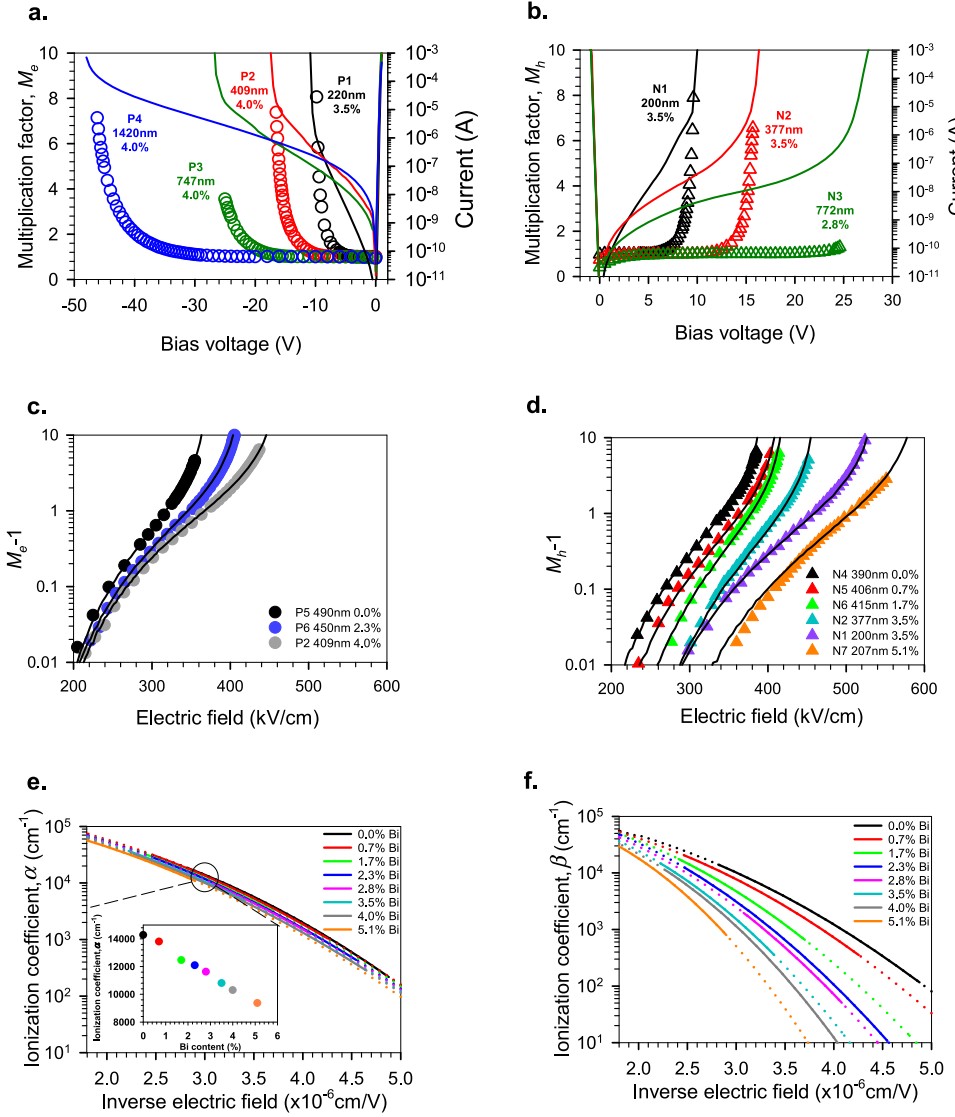

**Fig. 2 Photomultiplication characteristics and ionization coefficients in GaAsBi. a, b** The symbols show $M_e$ ($M_h$) versus applied reverse bias for p-i-n (n-i-p) structures of varying i-region thickness while the solid lines show the dark currents for 50−100 μm diameter devices. **c, d** $M_e$−1 ($M_h$−1) vs electric field for the 400 nm thick p-i-n (n-i-p) diodes of different Bi contents. The solid lines show the simulated results. **e** $\alpha$ vs inverse electric field at a range of Bi contents; inset shows $\alpha$ versus Bi content at an inverse electric field of $3 \times 10^{-6}$ cm/V. **f** $\beta$ of GaAsBi vs inverse electric field for a range of Bi contents. In (**e**, **f**) the dotted lines refer to the modelled $\alpha$ and $\beta$ values for the different Bi compositions; the solid lines indicate the values used to fit to the experimental data, as shown in (**c**, **d**).

would give $M_e$ values from 1.01 to 15 in 400 nm GaAsBi p-i-ns. As the Bi content increases and $\beta$ is reduced, the electric field necessary to achieve a given $M_e$ value increases. Even at high electric fields, the addition of 4.0 % Bi to GaAs more than doubles the $\alpha/\beta$ ratio.

To independently corroborate this increase in the $\alpha/\beta$ ratio, excess noise measurements as a function of gain were undertaken on the nominally 400 and 800 nm p-i-ns as shown in Fig. 3d. Details of the measurement system used to do this are given in the "Methods" and Supplementary sections. McIntyre[3] showed that the excess noise from electron-initiated multiplication, $F_e$, is dependent on the $\alpha/\beta$ ratio as:

$$F_e = kM_e + (1 - k)\left(2 - \frac{1}{M_e}\right) \qquad (3)$$

where $M_e$ is the electron-initiated multiplication and $k$ is defined as $\beta/\alpha$. For low excess noise, we need a large $\alpha/\beta$ ratio or small $k$. The results show a marked reduction in $F_e$ for the nominally 400

nm p-i-n structures with increasing Bi content, consistent with the increasingly large $\alpha/\beta$ ratios of the 2.3 % Bi (P6) and 4.0 % Bi (P2) devices. The effective $k$ value for GaAs of 0.48 is reduced to 0.3 (P6) and 0.2 (P2). Measurements on the thicker avalanching width structure with 4 % Bi (P4) could only be undertaken up to a multiplication of three due to high dark currents but nevertheless suggest that the $\alpha/\beta$ ratio increases further, giving $k \sim 0.15$ as the operating electric field is lowered. The effective $k$ value measured in the 400 nm thick structures is smaller than that which would be predicted by the $\alpha/\beta$ ratios in Fig. 3c. While the effect of the dead-space is not significant on the multiplication characteristics in avalanching widths that are 200 nm or larger, it does have a significant impact on the excess noise in structures thinner than about 600 nm[38], where it effectively reduces the measured $k$. Separating out the effect of a real $\alpha/\beta$ ratio increase due to the Bi from these dead-space effects requires the use of a non-local numerical model[35], knowledge of the electron and hole threshold energies ($E_{the}$ and $E_{thh}$ respectively) and the parameterized values

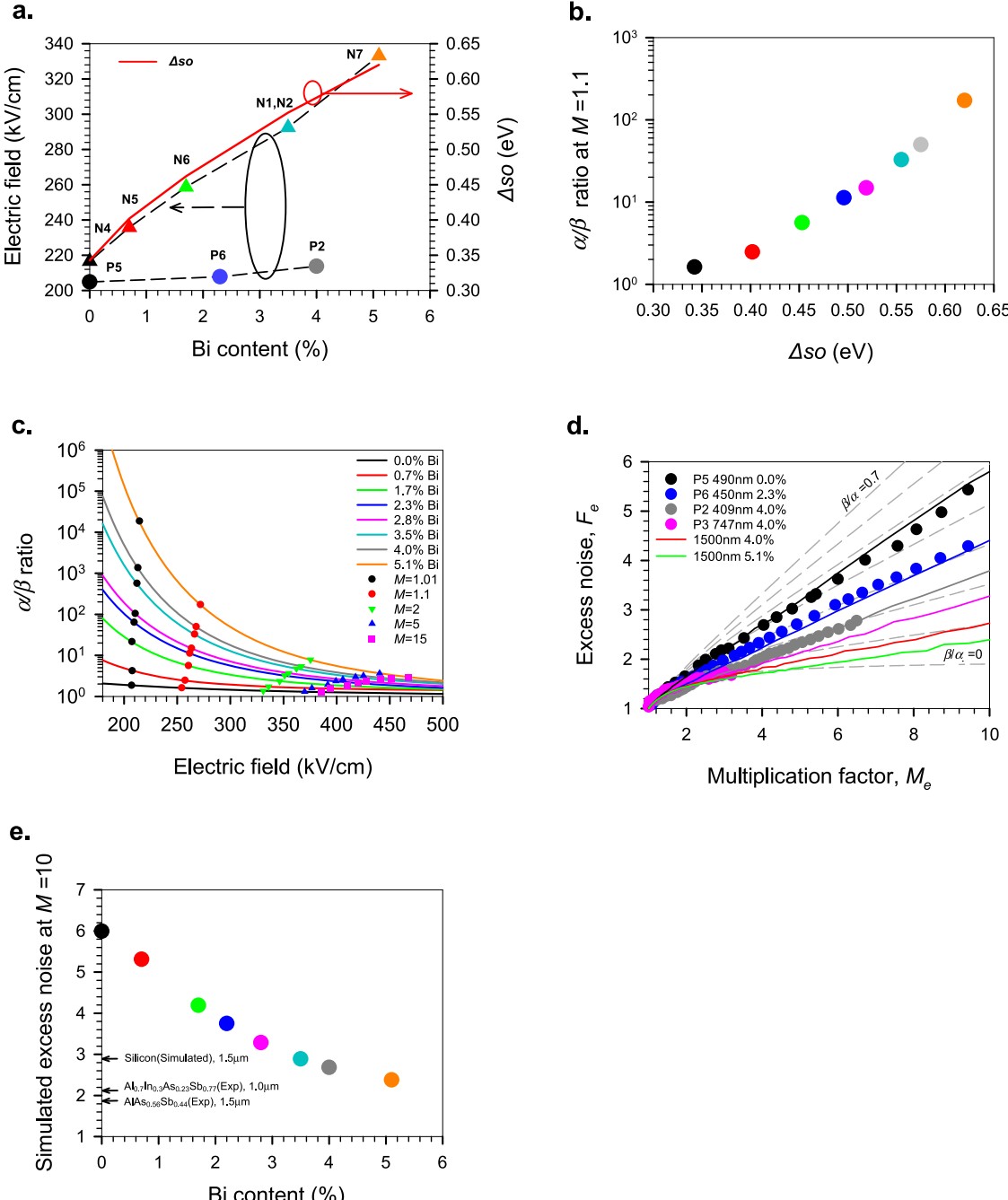

**Fig. 3 Effect of Bi on the $\alpha/\beta$ ratio of GaAs. a** The minimum electric field required to cause electrons and holes to ionize (defined as when $M_{e,h} = 1.01$) in nominally 400 nm GaAsBi p-i-ns and n-i-ps. The $\Delta so$ energy vs Bi content is also shown. **b** The $\alpha/\beta$ ratio (at $M_e = 1.1$) in GaAsBi as a function of the $\Delta so$ energy. **c** The calculated electric field dependence of the $\alpha/\beta$ ratio for different Bi contents. The symbols highlight the calculated $\alpha/\beta$ ratio at $M_e$ values of 1.01, 1.1, 2, 5, and 15 in 400 nm GaAsBi p-i-n diodes. **d** The measured excess noise, $F_e$ (symbols), in the nominally 400 nm and 800 nm thick p-i-n diodes with 4 % Bi, showing how adding Bi reduces the excess noise of GaAs. The solid lines are simulations that show good agreement with the measurements and predict that much lower excess noise is possible for 1500 nm thick structures with 4 % (red line) and 5.1 % (green line) Bi. The dashed lines represent the theoretical noise predictions by McIntyre[3] for $\beta/\alpha$ ratios from 0 to 0.7 in steps of 0.1. **e** The simulated excess noise ($F_e$) at $M = 10$ for 1500 nm thick avalanching structures with increasing Bi content. Also shown are the best reports for AlAsSb[40] and AlInAsSb[41], and simulations for silicon using data from Overstraeten and Man[5].

of $\alpha$ and $\beta$ from Fig. 2e, f to determine the avalanche multiplication excess noise. Implementing this model and using values of $E_{the} = 2.3$ eV and $E_{thh} = 2.1$ eV[39] gave good agreement with the measured excess noise not only for the GaAs structure, but also for the 2.3 % Bi and 4.0 % Bi structures as shown by the coloured lines in Fig. 3d. Operating at lower electric fields (via thicker avalanching region widths) or with higher Bi contents should give more significant excess noise reductions, and simulations in

Fig. 3d show that a 1500 nm p-i-n with 5.1 % Bi should have an $\alpha/\beta$ ratio of 20. Fig. 3e shows how the excess noise due to electron multiplication at a $M$ of 10 in 1500 nm thick avalanching structures is expected to decrease rapidly with increasing Bi% if the issue of high dark currents could be mitigated. This figure shows that while equivalent thickness Sb containing structures still have slightly lower excess noise[40,41], with the addition of just 3.5 % Bi to GaAs we expect the excess noise to be

comparable with silicon[5]. The $\Delta so$ energy increase with Bi is uniquely large, although appreciable increases may also be possible with the addition of antimony (Sb), another large group V atom. This mechanism might therefore be responsible, in part, for the large $\alpha/\beta$ ratios reported in[42], $Al_{0.85}Ga_{0.15}As_{0.56}Sb_{0.44}$[43], and $Al_xIn_{1-x}AsSb$[41] recently, where a significant fraction of Sb is incorporated.

In conclusion, we have experimentally demonstrated that increasing the spin-orbit splitting energy, $\Delta so$, in GaAs, by the addition of small amounts of bismuth, significantly reduces the hole ionization coefficient, while leaving the electron ionization coefficient largely unchanged. The results here clearly show that this approach of impeding the transfer of holes into the split-off band is one way to enhance the ionization coefficient ratio and reduce the excess noise in APDs. Adding large-atom group V elements to As-based III−V semiconductors provides a pathway to engineer the band-structure for hot-carrier processes.

## Methods

**Growth and diode fabrication**. The GaAsBi p-i-n and n-i-p structures (P1–P6 N1–N7) were grown on n- and p-GaAs (001) substrates respectively using an Omicron scanning tunnelling microscope-MBE system. Substrate heating under an As overpressure was used to remove the native surface oxide and Ga, Be, and Si fluxes were applied to deposit the cladding regions. The growth was paused for 20 min while the substrate temperature was dropped prior to GaAsBi deposition and the As flux changed from oversupplied $As_2$ to near-stoichiometry $As_4$. The low miscibility of GaBi and GaAs necessitates growth temperatures < 420 °C and a near stoichiometric III−V flux ratio; $As_4$ was used as it provides a larger growth window without detriment to the material quality[44]. Following GaAsBi growth, the substrate temperature was increased and the As flux changed again during another 20 min growth pause prior to the upper cladding layer growth. For details on the growth of each device, see the Supplementary MBE section. Circular mesa devices were fabricated with ohmic metal contacts using photolithography, metal evaporation, and wet chemical etching techniques developed for GaAs with no attempts made to passivate the surfaces.

**Dark current−voltage and capacitance−voltage measurements**. Dark current−voltage measurements were performed by using an HP4140B picoammeter while capacitance−voltage measurements were undertaken using an HP4275A LCR metre at a frequency of 1 MHz. The depletion width and background doping concentration were calculated by solving Poisson's equation with a static dielectric constant of 12.95.

**Photocurrent spectral response and XRD measurements**. Photocurrent and XRD were used to estimate the device bandgap energies and Bi contents. A Bruker D8 Discover X-ray diffractometer was used to perform $\omega$-$2\theta$ scans on each diode. The Bi contents were determined by fitting the XRD spectra using RADS Mercury software, in which the GaBi lattice constant was assumed to be 6.324 Å[45]. Room-temperature photocurrent measurements were undertaken on devices using a tungsten-lamp and a monochromator, and the bandgap energy was deduced from this. The Bi content determined in this manner showed very good agreement with the theoretical curve in Fig. 1c taking into account the strain relaxation (as detailed in the Supplementary section), with an uncertainty in Bi composition of ± 0.1 %. Absorption coefficients were extracted by determining the quantum efficiency of the GaAsBi samples in the range > 900 nm assuming that the GaAs cladding layers were effectively transparent.

**Multiplication and excess noise**. The multiplication, $M$, values as functions of reverse bias voltage were determined from the photocurrent measurements using a lock-in amplifier with phase-sensitive detection to remove the DC leakage currents. The use of 405 nm illumination, focussed onto the top optical windows, ensured that only electrons (holes) initiated the multiplication process in the p-i-ns (n-i-ps). Increases to the photocurrent due to a widening of the depletion widths with increasing bias were very small in all the samples investigated (see Supplementary section VIII) but were nevertheless corrected for using the methodology of Woods et al[46]. The excess noise, $F$, values as functions of $M$ were obtained using a specially designed circuit and with phase-sensitive detection techniques as described by Lau et al[47].

**Extraction of alpha and beta for different Bi content**. Wavelength-dependent multiplication measurements show that $\alpha > \beta$ in GaAsBi, and so using $M_e$ alone can give an initial accurate indication of $\alpha$ at low electric fields[48]. Using p-i-n diodes with similar Bi contents and different intrinsic region thicknesses enabled the behaviour of $\alpha$ to be determined over a wide range of electric field(s). With this and interpolating for the $\alpha$ of Bi compositions that were not experimentally available, an iterative technique was used to adjust the values of $\alpha$ and $\beta$ until good fits to

$M_e-1$ and $M_h-1$ were obtained for all the p-i-ns and n-i-ps shown in Table 1. Further details of this technique are given in the Supplementary section.

**Modelling of excess noise**. The parameterized electric field dependent values of $\alpha$ and $\beta$ from Fig. 2e, f were used to simulate the excess noise data in p-i-n diodes with different avalanche widths and Bi% using a numerical model[39]. The electron and hole ionization threshold energies were assumed to be similar to those of GaAs at 2.3 and 2.1 eV respectively.

## Data availability

The experimental and modelled data generated in this study have been deposited in the University of Sheffield Figshare database under https://doi.org/10.15131/shef.data.14816691.

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

## Acknowledgements

The authors are grateful to S. J. Sweeney (University of Surrey, UK) and C. A. Broderick (University College Cork, Ireland) for fruitful discussions on the band structure modelling, and D. Walker (X-ray Diffraction Research Technology Platform, University of Warwick, UK) for supplementary XRD data and assistance with the interpretation of the XRD results. We also like to acknowledge the assistance of F. Bastiman with the initial growth studies. This work has been supported by a Royal Academy of Engineering Research Fellowship Grant (RF1516\15\43) (R.D.R.), EPSRC Grant EP/N020715/1 (C.H.T. and J.P.R.D.), and by an EPSRC Doctoral Training Programme (T.B.O.R. and N.J.B.).

## Author contributions

Y.L., Z.Z., R.D.R., and J.P.R.D. designed the structures. Y.L., N.J.B., Z.Z., and T.B.O.R. carried out layer growth. N.J.B. and L.W.L. conducted the device fabrication. Y.L. undertook the experimental measurements. Y.L. and X.Y. undertook the multiplication modelling. Y.L., X.Y., R.D.R., C.H.T., and J.P.R.D. discussed and analyzed the results, and wrote the paper. J.P.R.D. and R.D.R. supervised the project. All authors reviewed the paper and approved the paper.

## Competing interests

The authors declare no competing interests.
