## [Peer Review File · Nature Communications]

Reviewers' Comments:

Reviewer #1:

Remarks to the Author:

This paper reported the GaAsBi material with very low excess noise, which is very important for APD application. The GaAsBi materials show more than 100-fold enhancement of alpha/beta ratio as Bi increases, which is very interesting. I recommend it to be published with some minor revisions.

1. In table 1, the authors seem doesn't have GaAsBi p-i-n with Bi up to 5.1%. However, the authors reported the modeled alpha value in Fig. 2(e). It is not clear how this model was done with no experimental data. I recommend the authors explain the model how you get alpha in GaAsBi with Bi of 5.1%; otherwise, it could be confusing.
2. What is intrinsic doping in the intrinsic region of GaAsBi p-i-n? In Fig S3, it seems to be above $1e16 \text{ cm}^{-3}$. How does the electrical field look like in the p-i-n or n-i-p? For formula 2(b) and 2(c), is that still ok to assume a constant electric field in the intrinsic region?
3. In Fig 2(e) and (f), the author reported the alpha and beta with different Bi; however, as shown in table 1, there are multiple samples with the same Bi composition, which sample is used to calculate the alpha/beta? Is the data based on the samples with the same Bi consistent with each other? I think the authors should also report that as well.

Reviewer #2:

Remarks to the Author:

This is an excellent paper. It has been suggested for several years that adding Bi to GaAs would increase the spin orbit splitting energy. It was anticipated that this would suppress impact ionization of holes, with the result of reducing the noise associated with the gain in an avalanche photodiode. However, to date, there has not been a good study of the behavior of Bi in GaAs-based avalanche photodiodes. The authors present a thorough study that should be of great interest to those who conduct research on avalanche photodiodes.

1. Figures 2 (a) and (b) show the dark current and gain of these APDs but there is no photocurrent curve. It is assumed that the photocurrent is constant at low bias, which enables accurate determination of the unity gain photocurrent, but it would be good to see the photocurrent versus bias.
2. The dark currents appear high, which may be understandable given the fact that the mesa sides were unpassivated. Have the authors measured the dark current versus diameter to determine if surface leakage dominates. Since some of the crystals are relaxed, it is possible that bulk defects are significant contributors to the dark current. It would be useful to know if the surface or bulk components of the dark current are most significant.

Response to Reviewer Comments (in red):

Reviewer #1 (Remarks to the Author):

This paper reported the GaAsBi material with very low excess noise, which is very important for APD application. The GaAsBi materials show more than 100-fold enhancement of alpha/beta ratio as Bi increases, which is very interesting. I recommend it to be published with some minor revisions.

We are pleased that the reviewer found this paper interesting.

1. In table 1, the authors seem doesn't have GaAsBi p-i-n with Bi up to 5.1%. However, the authors reported the modeled alpha value in Fig. 2(e). It is not clear how this model was done with no experimental data. I recommend the authors explain the model how you get alpha in GaAsBi with Bi of 5.1%; otherwise, it could be confusing.

The method used to extract the alpha and beta for GaAsBi of different compositions is explained in Supplementary section VIII. If we have M_e or M_h from a p-i-n or n-i-p respectively, the larger values of multiplication involve both alpha and beta as shown by equations 2b and 2c in the main paper. Although we do not have a 5.1% GaAsBi p-i-n, we have a n-i-p and so have M_h . Initially, we estimate the alpha from the M_e of the 2.3%, 3.5% and 4.0% p-i-ns, and the beta from the M_h of the 0.7%, 1.7%, 2.8%, 3.5% and 5.1% n-i-ps. Next, we interpolate (and extrapolate) to estimate the alpha and beta of compositions that we do not have, so we have an initial starting point for alpha and beta of all the different GaAsBi compositions investigated. We then adjust the exact values of these alpha and beta for the different compositions until we can fit to the experimentally measured $M_{e(h)}-1$ over a wide range of values. Although we do not have the alpha for the 5.1% GaAsBi, we can estimate its value such that it gives an excellent fit to the M_h-1 shown in Fig.2d of the main paper. Even a small 20% increase or decrease to this value of alpha changes the quality of the fit to the experimental data of the 5.1% GaAsBi layer as shown below. This means that when we have a good agreement between the experimental results and the simulated multiplication over a wide dynamic range, we can be confident of the accuracy of the ionization coefficients used. We have improved the description of the methodology used to extract alpha and beta in the Supplementary section VIII.

Fig. 1-1.

2. What is intrinsic doping in the intrinsic region of GaAsBi p-i-n? In Fig S3, it seems to be above $1e16 \text{ cm}^{-3}$. How does the electrical field look like in the p-i-n or n-i-p? For formula 2(b) and 2(c), is that still ok to assume a constant electric field in the intrinsic region?

The background doping is not 10^{16} cm^{-3} but more like $<10^{15} \text{ cm}^{-3}$. Figure 1-2a below shows the doping profile from C-V measurements including the 1600 nm p-i-n (P4). The first doping level we see for each device is at the depletion width due to the built-in voltage ($\sim 1.2 \text{ V}$) only. For this to deplete $\sim 1300 \text{ nm}$ in P4, the background doping has to be $< 5 \times 10^{14} \text{ cm}^{-3}$ as shown by a simple calculation of a $n^+ - p$ junction depletion width (Fig. 1-2b below). If the background doping was 10^{16} cm^{-3} we would only have $\sim 300 \text{ nm}$ of depletion at zero bias in P3, N3 and P4. With a low level of doping of $<10^{15} \text{ cm}^{-3}$ in the depletion region, we can safely assume that the electric-field is constant. To avoid any confusion, we now include the doping density of P4 in Fig. S3 in the Supplementary section and make clear that the background doping in the intrinsic region is $< 10^{15} \text{ cm}^{-3}$.

Fig. 1-2a.

Fig. 1-2b.

3. In Fig 2(e) and (f), the author reported the alpha and beta with different Bi; however, as shown in table 1, there are multiple samples with the same Bi composition, which sample is used to calculate the alpha/beta? Is the data based on the samples with the same Bi consistent with each other? I think the authors

should also report that as well.

Samples with identical Bi% shown in Table 1 (e.g. P2, P3, P4 or N1, N2) have different avalanche widths. What this means is that they will cover slightly different electric field ranges, with the thinner structures covering a higher electric field. Where the electric field ranges overlap, the alpha (beta) values will be identical. In obtaining the alpha and beta shown in Fig. 2e,f, all the samples were used to either give values for different Bi%, or to cover a wider electric field range for the same Bi%.

Reviewer #2 (Remarks to the Author):

This is an excellent paper. It has been suggested for several years that adding Bi to GaAs would increase the spin orbit splitting energy. It was anticipated that this would suppress impact ionization of holes, with the result of reducing the noise associated with the gain in an avalanche photodiode. However, to date, there has not been a good study of the behavior of Bi in GaAs-based avalanche photodiodes. The authors present a thorough study that should be of great interest to those who conduct research on avalanche photodiodes.

We thank the reviewer for his generous comments.

1. Figures 2 (a) and (b) show the dark current and gain of these APDs but there is no photocurrent curve. It is assumed that the photocurrent is constant at low bias, which enables accurate determination of the unity gain photocurrent, but it would be good to see the photocurrent versus bias.

Figs 2-1a and 2-1b below shows examples of the actual photocurrent (as measured by the lock-in amplifier) versus reverse bias for the devices investigated. Measurements were taken on different devices on each wafer and with different laser intensities to ensure reproducibility of results. The reviewer is correct in assuming that the photocurrent is almost constant at low biases prior to the onset of avalanche multiplication. We show this more clearly in Figs. 2.1c and 2.1d where the photocurrents are plotted on a linear scale and normalised. In reality, there is a very small increase in this photocurrent with reverse bias which is imperceptible unless the figure is greatly expanded. This was corrected for in determining the ionization coefficients as detailed in the Methods section. We now add this figure to section VIII of the Supplementary section for completeness. The Methods section in the main paper now makes clear that the photocurrent is almost constant and refers to the Supplementary section for the full data.

Fig. 2-1a.

Fig. 2-1b.

Fig. 2-1c.

Fig. 2-1d.

2. The dark currents appear high, which may be understandable given the fact that the mesa sides were unpassivated. Have the authors measured the dark current versus diameter to determine if surface leakage dominates. Since some of the crystals are relaxed, it is possible that bulk defects are significant contributors to the dark current. It would be useful to know if the surface or bulk components of the dark current are most significant.

In the main paper we state that the forward currents shown in Fig 2a,b (main paper) scale with area, so they are bulk currents, rather than perimeter dependent. The reverse dark currents (examples shown in Fig. 2.2a-c below for three layers and different size radii) are more variable and do not scale well with either area or perimeter. They may be very approximately following a combination of 'bulk' like behaviour (due to Bi defect states and/or defects due to strain relaxation) and surface leakage. A statement to the effect that both mechanisms are contributing to the dark currents is made in the main paper, just before Fig.2.

Fig. 2-2a.

Fig. 2-2b.

Fig. 2-2c.

Reviewers' Comments:

Reviewer #1:

Remarks to the Author:

I think the authors have addressed all the points, and it should be published in nature comm.

Reviewer #2:

Remarks to the Author:

I think the paper is now acceptable for publication.